

# The BACKMAP Python module: how a simpler Ramachandran number can simplify the life of a protein simulator

Ranjan Mannige

The Multiscale Institute, Berkeley Lake, GA, USA

## ABSTRACT

Protein backbones occupy diverse conformations, but compact metrics to describe such conformations and transitions between them have been missing. This report re-introduces the Ramachandran number ($\mathcal{R}$) as a residue-level structural metric that could simply the life of anyone contending with large numbers of protein backbone conformations (e.g., ensembles from NMR and trajectories from simulations). Previously, the Ramachandran number ($\mathcal{R}$) was introduced using a complicated closed form, which made the Ramachandran number difficult to implement. This report discusses a much simpler closed form of $\mathcal{R}$ that makes it much easier to calculate, thereby making it easy to implement. Additionally, this report discusses how $\mathcal{R}$ dramatically reduces the dimensionality of the protein backbone, thereby making it ideal for simultaneously interrogating large numbers of protein structures. For example, 200 distinct conformations can easily be described in one graphic using $\mathcal{R}$ (rather than 200 distinct Ramachandran plots). Finally, a new Python-based backbone analysis tool—BACKMAP—is introduced, which reiterates how $\mathcal{R}$ can be used as a simple and succinct descriptor of protein backbones and their dynamics.

## INTRODUCTION

Proteins are a class of biomolecules unparalleled in their functionality (*Berg, Tymoczko & Stryer, 2010*). A natural protein may be thought of as a linear chain of amino acids, each normally sourced from a repertoire of 20 naturally occurring amino acids. Proteins are important partially because of the structures that they access: the conformations (conformational ensemble) that a protein assumes determines the functions available to that protein. However, all proteins are dynamic: even stable proteins undergo long-range motions in their equilibrium states; that is, they have substantial diversity in their conformational ensemble (*James & Tawfik, 2003a*, *2003b*; *Oldfield et al., 2005*; *Tokuriki & Tawfik, 2009*; *Schad, Tompa & Hegyi, 2011*; *Vertessy & Orosz, 2011*; *Mannige, 2014*). Additionally, a number of proteins undergo conformational transitions, without which they may not properly function. Finally, some proteins—intrinsically disordered proteins— display massive disorder with conformations that dramatically change over time

Corresponding author
Ranjan Mannige,
ranjanmannige@gmail.com

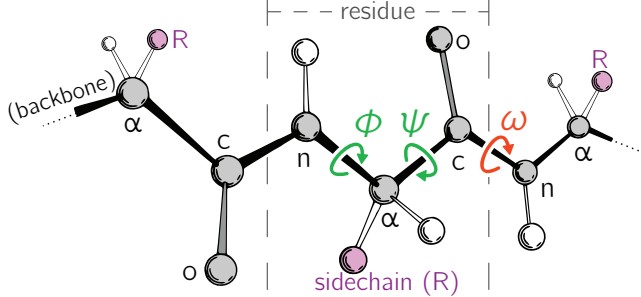

**Figure 1** **Degrees of freedom available to a protein backbone.** Backbone conformational degrees of freedom dominantly depend on the dihedral angles $\phi$ and $\psi$ (green), and to a smaller degree depend on the third dihedral angle ($\omega$; red) as well as bond lengths and angles (unmarked).

(*Uversky, 2003*; *Fink, 2005*; *Midic et al., 2009*; *Espinoza-Fonseca, 2009*; *Uversky & Dunker, 2010*; *Tompa, 2011*; *Sibille & Bernado, 2012*; *Kosol et al., 2013*; *Dunker et al., 2013*; *Geist et al., 2013*; *Baruah, Rani & Biswas, 2015*), and whose characteristic structures are still not well-understood (*Beck et al., 2008*).

Large-scale changes in a protein occur due to changes in protein backbone conformations. Figure 1 is a cartoon representation of a peptide/protein backbone, with the backbone bonds themselves represented by darkly shaded bonds. *Ramachandran, Ramakrishnan & Sasisekharan (1963)* had recognized that the backbone conformational degrees of freedom available to an amino acid (residue) $i$ can be almost completely described by only two dihedral angles: $\phi_i$ and $\psi_i$ (Fig. 1, green arrows). Today, Ramachandran plots are used to qualitatively describe protein backbone conformations.

The Ramachandran plot is recognized as a powerful tool for two reasons: (1) it serves as a map for structural "correctness" (*Laskowski et al., 1993*; *Hooft, Sander & Vriend, 1997*; *Laskowski, 2003*), since many regions within the Ramachandran plot space are energetically not permitted (*Momen et al., 2017*); and (2) it provides a qualitative snapshot of the structure of a protein (*Berg, Tymoczko & Stryer, 2010*; *Alberts et al., 2002*; *Subramanian, 2001*; *Lovell et al., 2003*). For example, particular regions within the Ramachandran plot indicate the presence of particular locally-ordered secondary structures such as the α-helix and β-sheet (see Fig. 2A).

While the Ramachandran plot has been useful as a measure of protein backbone conformation, it is not popularly used to assess structural dynamism and transitions (unless specific knowledge exists about whether a particular residue is believed to undergo a particular structural transition). This is because of the two-dimensionality of the plot: describing the behavior of every residue involves tracking its position in two-dimensional ($\phi$, $\psi$) space. For example, a naive description of positions of a peptide in a Ramachandran plot (Fig. 2B) needs more annotations for a per-residue analysis of the peptide backbone's structure. Given enough residues, it would be impractical to track the position of each residue within a plot. This is compounded by time, as each point in (Fig. 2B) becomes a curve in (Fig. 2C), further confounding the situation. The possibility of picking out previously unseen conformational transitions and dynamism becomes a
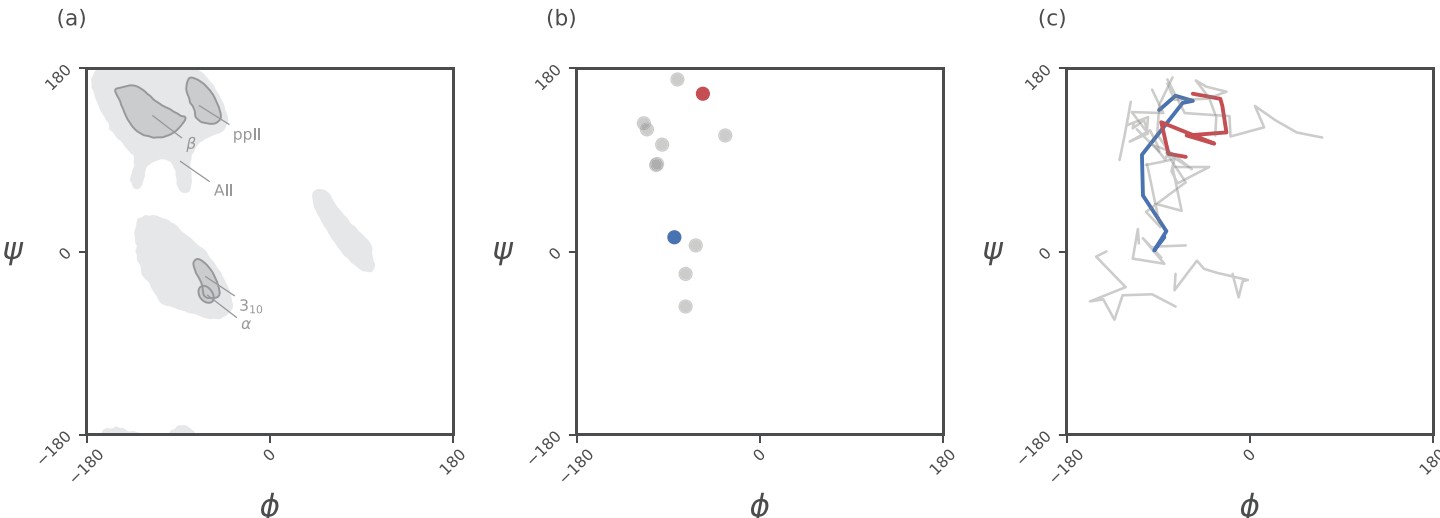

**Figure 2 Ramachandran plots allow for the per-residue representation of backbone conformation.** (A) represents regions in the plot that are occupied by backbones describing particular regular secondary structures. (B) represents the positions of a 12 residue peptide, with one circle per residue. (C) represents a trajectory within which the backbone conformation is allowed to change. Here, each curve represents the evolution of each residue, with corners representing discrete states. While the Ramachandran plot is useful for getting a *qualitative* sense of peptide backbone structure (A, B), it is not a convenient representation for exploring peptide backbone dynamics (C). Secondary structure keys used here and throughout the document: α, α-helix; $3_{10}$, $3_{10}$-helix; β, β-sheet/extension; ppII, polyproline II helix.   

logistical impracticality. As indicated above, this impracticality arises primarily from the fact that the Ramachandran plot is a two-dimensional map.

Beyond Ramachandran plots, tracking changes in a protein trajectory is either overly detailed or overly holistic: an example of an overly detailed study is the tracking of exactly one or a few atoms over time (this already poses a problem, since we would need to know exactly which atoms are expected to partake in a transition); an example of a holistic metric is the radius of gyration (this also poses a problem, since we will never know which residues contribute to a change in radius of gyration without additional interrogation). With our understanding of protein dynamics undergoing a new renaissance—especially due to intrinsically disordered proteins and allostery—having hypothesis-agnostic yet detailed (residue-level) metrics of protein structure has become even more relevant. But unfortunately, there has been no single compact descriptor of protein structure. This impedes the naïve or hypothesis-free exploration of new trajectories/ensembles.

It has recently been shown that the two Ramachandran backbone parameters ($\phi$, $\psi$) may be conveniently combined into a single number—the Ramachandran *number* [$\mathcal{R}(\phi, \psi)$ or simply $\mathcal{R}$]—with little loss of information (Fig. 3; *Mannige, Kundu & Whitelam, 2016*). In a previous report, detailed discussions were provided regarding the reasons behind and derivation of $\mathcal{R}$ (*Mannige, Kundu & Whitelam, 2016*). This report provides a simpler version of the equation previously published (*Mannige, Kundu & Whitelam, 2016*), and further discusses how $\mathcal{R}$ may be used to provide information about protein ensembles and trajectories. Finally, this report introduces a software package—BACKMAP—that can be used to produce pictograms that describe the
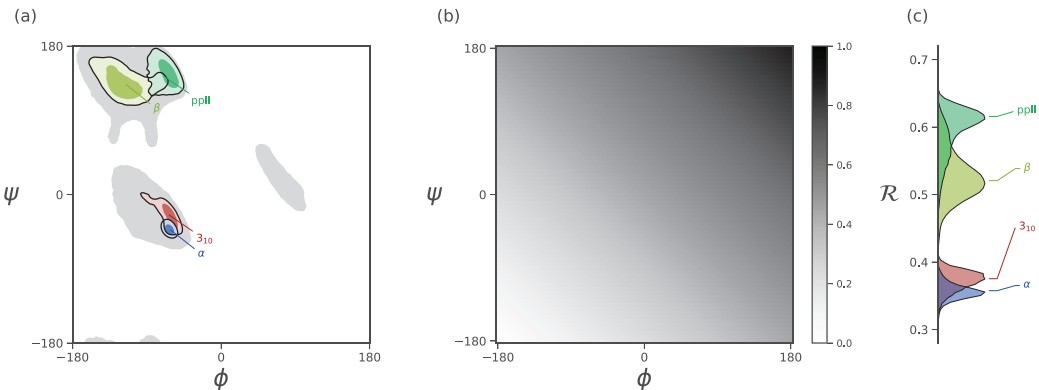

**Figure 3 Mapping of $\phi$ and $\psi$ to $\mathcal{R}$.** (A) describes the distribution of dominant regular secondary structures. (B) shows the mapping between the $(\phi, \psi)$ and $\mathcal{R}$. In particular, $\mathcal{R}$ increases in negative-sloping sweeps from the bottom left to the top right of the Ramachandran plot. (C) describes the distribution of secondary structures in $\mathcal{R}$ space. Both Ramachandran plots (A) and Ramachandran "lines" (C) equally resolve the secondary structure space, thereby making $\mathcal{R}$ a compact yet faithful representation of backbone structure (*Mannige, Kundu & Whitelam, 2016*).

behavior of a protein backbone within user-inputted conformations, structural ensembles, and trajectories. Given that each pictogram provides a picture of the whole protein backbone (i.e., all $\phi$ and $\psi$ angles), these pictograms are named multi-angle pictures (MAPs). BACKMAP is presently available on GitHub (https://github.com/ranjanmannige/BackMAP).

# INTRODUCING THE *SIMPLIFIED* RAMACHANDRAN NUMBER ($\mathcal{R}$)

The Ramachandran number is both an idea and an equation. Conceptually, the Ramachandran number ($\mathcal{R}$) is any closed form that collapses the dihedral angles $\phi$ and $\psi$ into one structurally meaningful number (*Mannige, Kundu & Whitelam, 2016*). *Mannige, Kundu & Whitelam (2016)* presented a version of the Ramachandran number (shown in the appendix as Eq. (7)) that was complicated in closed form, thereby reducing its utility. Here, a simpler and more accurate version of the Ramachandran number is introduced. The appendix shows how this simplified form was derived from the original closed form (Eq. (7)).

Given arbitrary limits of $\phi \in [\phi_{\min}, \phi_{\max})$ and $\psi \in [\psi_{\min}, \psi_{\max})$, where the minimum and maximum values differ by 360°, the most general and accurate equation for the Ramachandran number is

$$\mathcal{R}(\phi, \psi) \equiv \frac{\phi + \psi - (\phi_{\min} + \psi_{\min})}{(\phi_{\max} + \psi_{\max}) - (\phi_{\min} + \psi_{\min})}. \tag{1}$$

For consistency, we maintain throughout this paper that $\phi_{\min} = \psi_{\min} = -180°$ or $-\pi$ radians, which makes

$$\mathcal{R}(\phi, \psi) = \frac{\phi + \psi + 2\pi}{4\pi}. \tag{2}$$

 

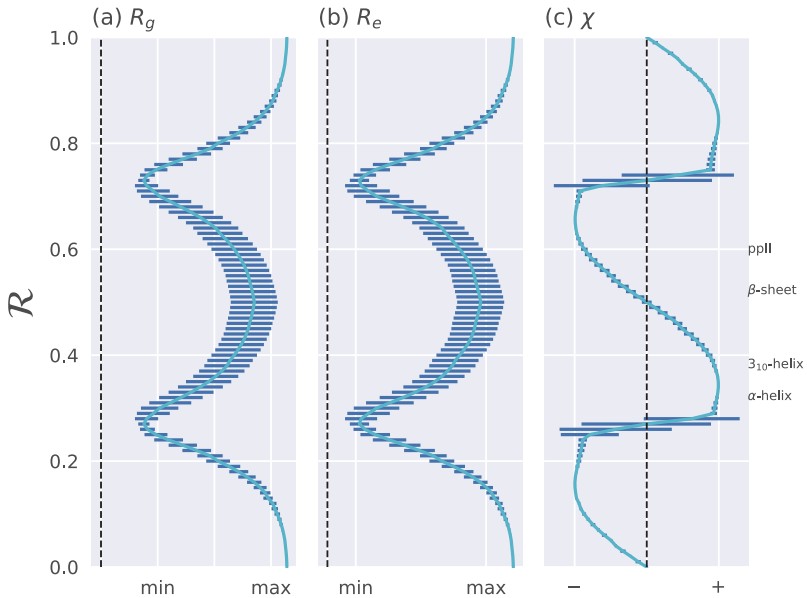

**Figure 4** **Relationships between $\mathcal{R}$ and other structural features.** The Ramachandran number $\mathcal{R}$ displays smooth relationships with respect to radius of gyration ($R_g$; A), end-to-end distance ($R_e$; B), and chirality ($\chi$; C), as calculated within *Mannige (2017)*. Light blue lines are average trends, dark blue horizontal lines are error bars. Average positions of dominant secondary structures are shown to the right. These trends explain why $\mathcal{R}$ is a useful and compact structural measure. Structural measures $R_g$, $R_e$, and $\chi$ were obtained by computationally generating polyglycine peptides of length 10 for all possible $\phi$ and $\psi \in [-180, -175, \ldots, 175, 180]$. This was done using the Python library PeptideBuilder (*Tien et al., 2013*). Values for $R_g$, $R_e$, and $\chi$ were obtained for each peptide and binned with respect to its $\mathcal{R}(\phi, \psi)$ (each bin represents a region in $\mathcal{R}$ space that is 0.01 $\mathcal{R}$ in width). Given that actual values for $R_g$ and $R_e$ mean little (since one rarely deals with polyglycines of length 10), actual values are omitted. $\chi$ ranges from $-1$ to $+1$.

As evident in Fig. 3, the secondary structure distributions within the Ramachandran plot are faithfully reflected in corresponding distributions within Ramachandran number space. This paper shows how the Ramachandran number is both compact enough and informative enough to generate immediately useful graphs (MAPs) of a dynamic protein backbone.

# REASONS TO USE THE RAMACHANDRAN NUMBER

## Ramachandran numbers are structurally meaningful

In addition to resolving positions of secondary structures (Fig. 3), $\mathcal{R}$ corresponds well to structural measures such as radius of gyration ($R_g$), end-to-end distance ($R_e$), and chirality ($\chi$). These relationships are shown in Fig. 4. Note that chirality comes in many forms, for example, one could be talking about different stereo-isomers, such as L vs D amino acids, or one could be concerned with left-twisting versus right-twisting backbones, that is, handedness (*Mannige, 2017*). This report will primarily be focused on chirality in context of backbone twist/handedness.

The trends in Fig. 4 show that as one progresses from low to high $\mathcal{R}$, various structural properties also progress smoothly. Additionally, backbones that display similar $\mathcal{R}$ also show little variation in structural properties, as evidenced by the small standard deviation bars. It is also important to note that the standard deviations shown in Fig. 4 were calculated

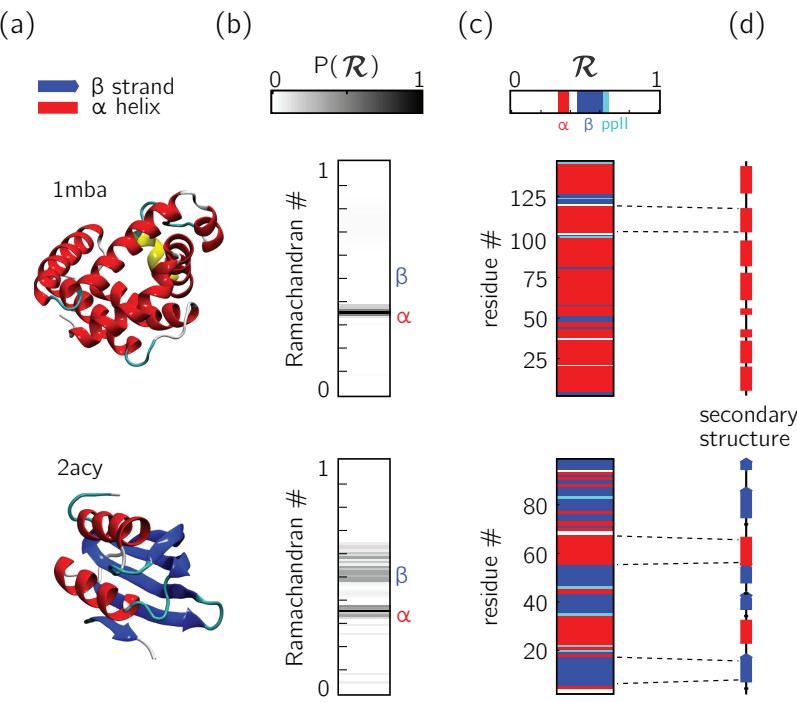

**Figure 5** **Two types of $\mathcal{R}$-codes.** Digesting protein structures (A) using $\mathcal{R}$ numbers either as histograms (B) or per-residue codes (C) allow for compact representations of salient structural features. For example, a single glance at the histograms indicate that protein 1mba is likely all α-helical, while 2acy is likely a mix of α-helices and β-sheets. Additionally, residue-specific codes (C) not only indicate secondary structure content, but also secondary structure stretches (compare to D), which gives a more complete picture of how the protein is linearly arranged.

by first populating every possible region of $(\phi, \psi)$-space. However, in reality, most regions of $(\phi, \psi)$-space are unoccupied due to steric/electrostatic constraints, which means that these error bars are likely to be even smaller for natural protein backbones than those depicted here. Finally, the $\mathcal{R}$ number is calculated by taking "sweeps" of the $(\phi, \psi)$-space in lines that are parallel to the negatively-sloping diagonal. Interestingly, such "sweeps" encounter only one major (dense) region within $(\phi, \psi)$-space (e.g., $\mathcal{R}$'s in the general vicinity of 0.34 represent structures that resemble α-helices). This means that $\mathcal{R}$ can also be used to assess the types of secondary structure present in a protein conformation.

## Ramachandran codes are stackable

An important aspect of the Ramachandran number ($\mathcal{R}$) lies in its compactness compared to the traditional Ramachandran pair $(\phi, \psi)$. The value of the conversion from $(\phi, \psi)$-space to $\mathcal{R}$-space is that the structure of a protein can be described in various one-dimensional arrays (per-structure "Ramachandran codes" or "$\mathcal{R}$-codes" or multi-angle maps); see, e.g., Fig. 5.

In addition to assuming a small form factor, $\mathcal{R}$-codes may then be *stacked* side-by-side for visual and computational analysis. There lies its true power.

For example, the one-$\mathcal{R}$-to-one-residue mapping means that the entire residue-by-residue structure of a protein can be shown using a string of $\mathcal{R}_i$s (which would show regions of

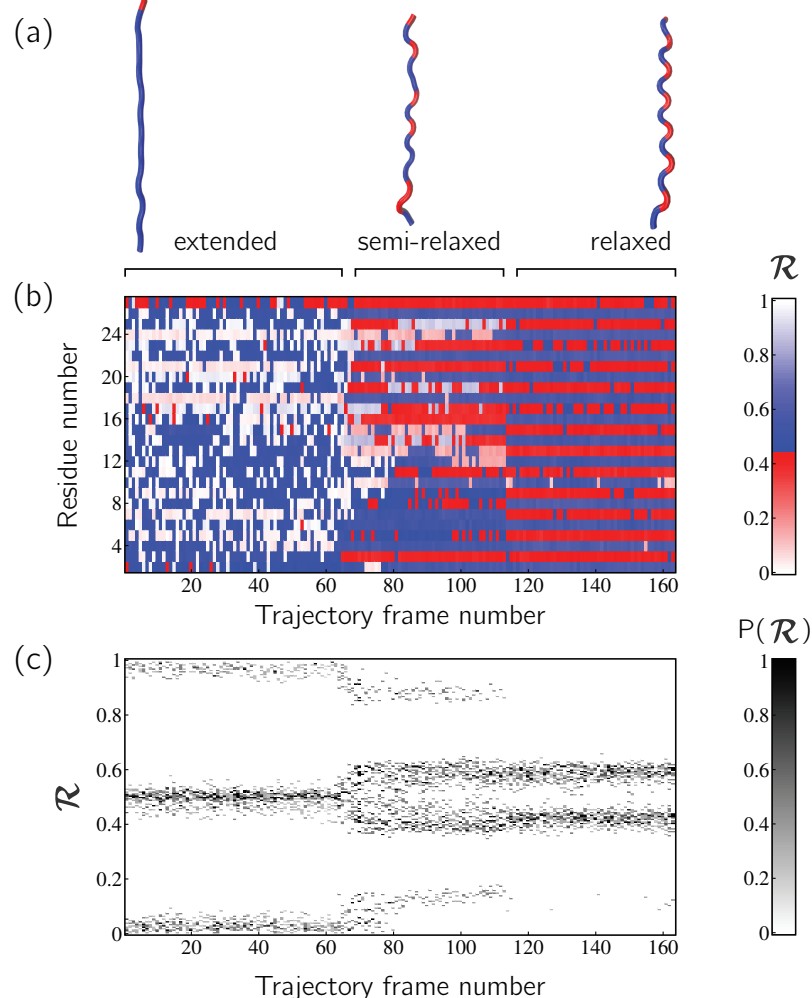

Figure 6 Stacked $\mathcal{R}$-codes provide useful information at a glance. Each panel represents a molecular dynamics simulation of a peptoid nanosheet (*Mannige, Kundu & Whitelam, 2016*), where each peptoid backbone was held (energetically restrained) in extended state in the beginning, upon which each backbone was allowed to relax by lifting the restrains. (A) displays representative structures from each stage of the simulation. (B) represents how the per-residue structure of the peptide evolved over "time" (the progression of time is represented as increasing frame number). (C) represents how the general distribution of backbone conformations in the peptoid (as evident by the $\mathcal{R}$ histogram) evolves over time.

secondary structure and disorder, for starters). Additionally, an entire protein's backbone makeup can be shown as a histogram in $\mathcal{R}$-space (which may reveal a protein's topology). The power of this format lies not only in the capacity to distill complex structure into compact spaces, but in its capacity to display *many* complex structures in this format, side-by-side (stacking).

Peptoid nanosheets (*Mannige et al., 2015*) will be used here as an example of how multiple structures, in the form of $\mathcal{R}$-codes, may be stacked to provide immediately useful pictograms. Peptoids are stereo-isomers of peptides, where the sidechain is attached to the backbone nitrogen rather than the carbon atom. Since both peptoids and peptides

(a)

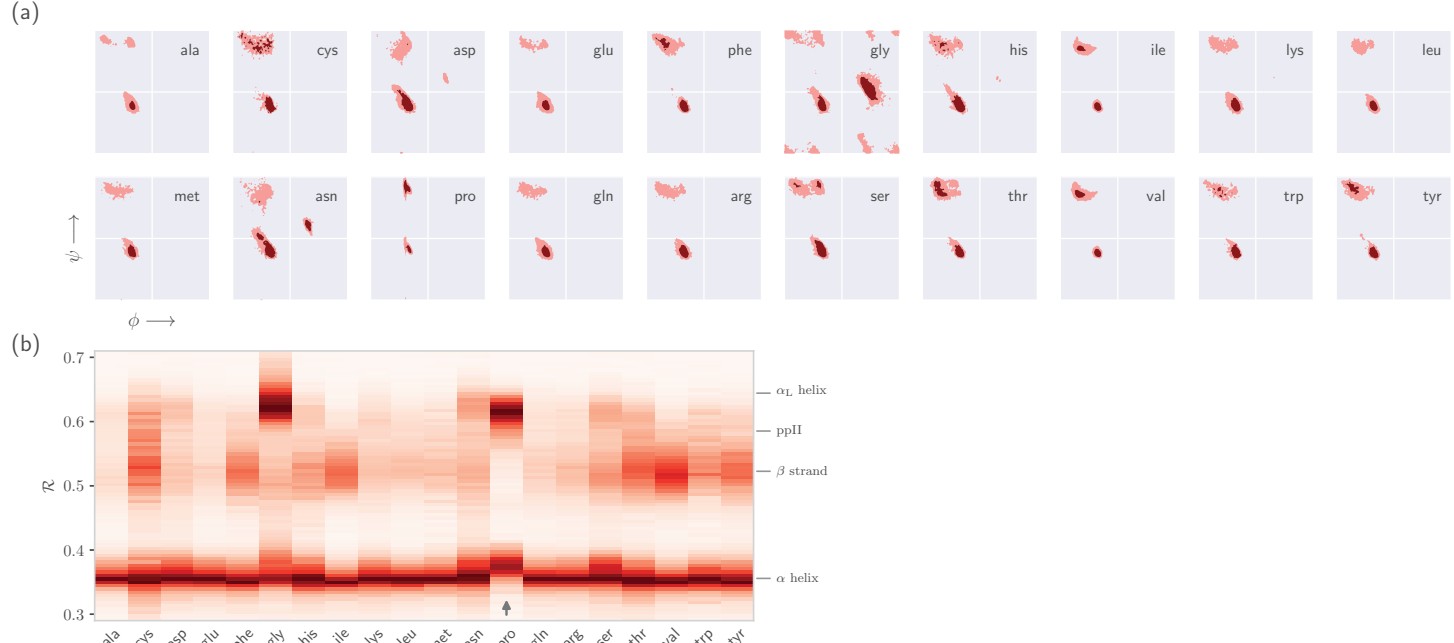

(b)

Figure 7 **Combining Ramachandran plots for all amino acids into one graph.** (A) shows the backbone behavior of each amino acid (three-letter codes used) found in the protein databank (see Methods). While these plots are instructive, it is difficult to compare such plots. For example, it is difficult to pick out the change in the α-helial region of the proline plot (pro). However, when we convert Ramachandran plots to Ramachandran lines [by converting $(\phi_i, \psi_i) \rightarrow \mathcal{R}_i$], we are able to conveniently "stack" Ramachandran lines calculated for each residue (B). Then, even visually, it becomes obvious that proline does not occupy the canonical α-helix region, which is not evident to an untrained eye in (A).

share identical backbone connectivity, the analysis described below could be applied to both peptides and peptoids.

Peptoid nanosheets are a recently discovered peptide-mimic that, in one molecular dynamics simulation (*Mannige et al., 2015*), were shown to display a novel secondary structure. In the reported model (*Mannige et al., 2015*), each peptoid within the nanosheet displays backbone conformations that alternate in chirality, causing the backbone to look like a meandering snake that nonetheless maintains an overall linear direction. This secondary structure was discovered by first setting up a nanosheet where all peptoid backbones were restrained to be fully extended, after which the restraints were energetically softened and completely released. Figure 6A shows snapshots of each of these states. As evident in Figs. 6B and 6C, the two types of $\mathcal{R}$-code stacks display salient information at first glance: (1) Fig. 6B shows that the extended backbone first undergoes some rearrangement with softer restraints, and then becomes more binary in arrangement as we look down the backbone; and (2) Fig. 6C shows that lifting restraints on the backbone causes a dramatic change in backbone topology, namely a birth of a bimodal distribution evident in the two parallel horizontal bands.

By utilizing $\mathcal{R}$, maps such as those in Fig. 6 provide information about every $\phi$ and $\psi$ within the backbone. As such, these maps are dubbed MAPs. A Python package called BACKMAP created Figs. 6A and 6B, and is provided as a GitHub repository (https://github.com/ranjanmannige/BackMAP). BACKMAP takes in a PDB

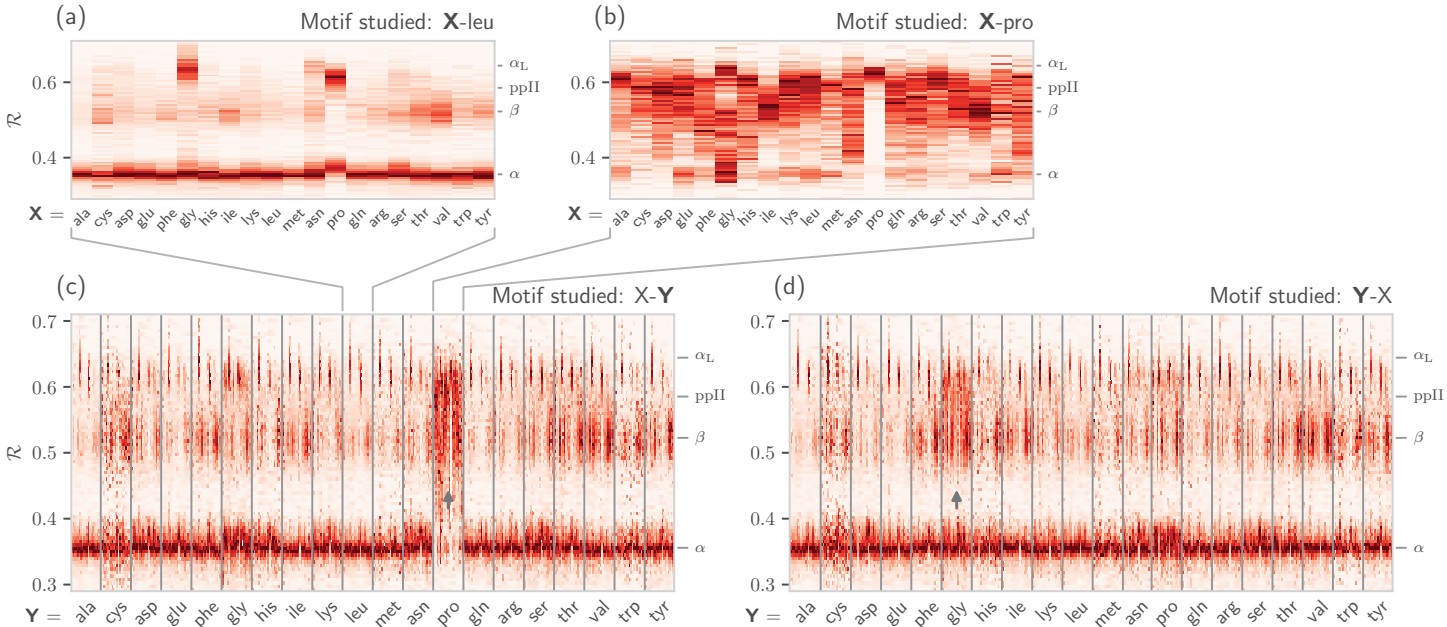

**Figure 8 How residue neighbors modify structure.** Similar to Fig. 7B, (A) represents the behavior of an amino acid "**X**" situated before a leucine (**X**-leu; assuming that we are reading a sequence from the N terminal to the C terminal). (B) similarly represents the behavior of specific amino acids situated before a proline (**X**-pro). While residues preceding a leucine behave similarly to their average behavior (Fig. 7A), most residues preceding prolines appear to be enriched in structures that change "direction" or backbone chirality (this is evident by many amino acids switching from $\mathcal{R} < 0.5$ to $\mathcal{R} > 0.5$). (C) and (D) show the behavior of individual amino acids when situated before and after each of the 20 amino acids, respectively. (C) and (D) show a major benefit of side-by-side Ramachandran line "stacking": general trends become much more obvious. For example, it is evident that prolines dramatically modify the structure of an amino acid preceding it (compared to average behavior of amino acids in Fig. 7B), while residues following glycines also have a higher prevalence of $\mathcal{R} > 0.5$ conformations (both trends are indicated by small arrows). Such trends, while previously discovered (see text), would not be accessible when naïvely considering Ramachandran plots because one would require 400 (20 × 20) distinct Ramachandran plots to compare. Note that the statistics for each $\mathcal{R}$-line in (C) and (D) are dependent on the joint prevalence of the residues being considered. For this reason, some $\mathcal{R}$-lines (e.g., those associated with cysteines) look more rough or "dotty" than others.

structure file containing a single structure, or multiple structures separated by the code "MODEL."

## Case study: picking out subtle differences from high volume of data

This section expands on the notion that $\mathcal{R}$-numbers—due to their compactness/stackability—can be used to pick out backbone structural trends that would be hard to decipher using any other metric. For example, it is well known that prolines (pro) display unusual backbone behavior: in particular, proline backbones occupy structures that are close to but distinct from α-helical regions. Due to the two-dimensionality of Ramachandran plots (Fig. 7A), such distinctions are hard to visually pick out from Ramachandran plots. However, stacking per-amino-acid $\mathcal{R}$-codes side-by-side make such differences patent (Fig. 7B; see arrow).

It is also known that amino acids preceeding prolines display unusual shifts in backbone twist/chirality. For example, Fig. 8C shows that amino acids appearing before prolines behave differently than they would otherwise (see the upward-facing arrow). Additionally, amino acids *following* glycines also appear to have their structures modified

(Fig. 8D; upward arrow). Note that these results are not new, and it has already been confirmed that, for example, nearest neighbors affect the conformational behavior of an amino acid as witnessed within Ramachandran plots (*Ting et al., 2010*), and proline changes the backbone conformation of the preceeding residue (*Gunasekaran et al., 1998*; *Ho & Brasseur, 2005*). However, Figs. 7 and 8 indicate that such information can be more concisely shown/identified when structures are stacked side-by-side in the form of $\mathcal{R}$-codes. Such subtle changes are often witnessed when protein backbones transition from one state to another.

## USING THE BACKMAP PYTHON MODULE

### Installation

BACKMAP may either be installed locally by downloading the GitHub repository, or installed directly by running the following line in the command prompt (assuming that pip exists): `> pip install backmap`.

### Usage

The module can either be imported and used within existing scripts, or used as a standalone package using the command "python -m backmap." First the in-script usage will be discussed.

#### In-script usage I: first simple test

The simplest test would be to generate Ramachandran numbers from $(\phi, \psi)$ pairs:

```
# Import module                                                        1
import backmap                                                         2
# Convert (phi, psi) to R                                              3
print backmap.R(phi =0, phi =0) # Expected output : 0.5                4
print backmap.R(  −180,   −180) # Expected output : 0.0                5
print backmap.R(   180,    180) # Expected output : 1.0 (equivalent in meaning to 0) 6
```

#### In-script usage II: basic usage for creating multi-angle pictures

The following code shows how MAPs of protein backbones can be generated:

**1. Select and read a protein PDB file**

Each trajectory frame must be a set of legitimate protein databank (PDB) "ATOM" records separated by "MODEL" keywords (distinct models show up as distinct frames on the $x$-axis or abscissa).

```
import backmap                                                                        1
import matplotlib.pyplot as plt     # Pyplot is needed for drawing graphs             2
import numpy as np                  # Numpy is needed for data transformations        3
pdbfn = './tests/pdbs/1xqq.pdb'     # Set pdb name                                     4
data = backmap.read_pdb(pdbfn)      # READ PDB in the form of a matrix with columns    5
```

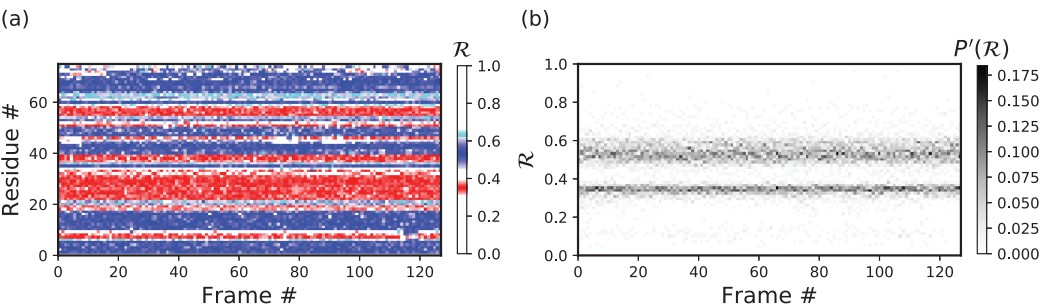

**Figure 9 In-script examples.** Two examples of pictures generated using code in sections "In-script usage II" (A) and "Inscript usage III." (A) and (B), respectively, represent a residue-level and histogram-level representation of backbone structure.

Here, "`data`" is a 2d array with four columns ["`model`," "`chain`," "`resid`," "R"]. The first row of "`data`" is the header, with values that follow.

## 2. Select color scheme (color map)

In addition to custom colormaps listed in the next section, one can also use traditional colormaps available at matplotlib.org (e.g., "`Reds`" or "`Reds_r`").

```
# setting the name of the colormap                                        6
cmap = "SecondaryStructure"                                               7
```

## 3. Draw per-chain MAPs

```
# Grouping by chain                                                       8
grouped_data = backmap.group_data_by (data, group_by='chain',             9
                      columns_to_return=['model','resid','R'])           10
for chain in grouped_data.keys(): # Going through each chain             11
    # Getting the X,Y,Z values for each entry                            12
    models, residues, Rs = grouped_data [chain]                          13
    # Finally, creating (but not showing) the graph                      14
    backmap.draw_xyz(X = models    ,         Y = residues    ,    Z = Rs 15
            , xlabel  = 'Frame #', ylabel   = "Residue #",zlabel = 'R'   16
              ,cmap = cmap     ,    title    = "Chain: ' "+chain+" ' "   17
              ,vmin=0,vmax=1)                                            18
    # Now, we display the graph:                                         19
    plt.show() # ... one can also use plt.savefig() to save to file      20
```

The code above results in Fig. 9A.

Additionally, by changing how one assigns values to "X" and "Y," one can easily construct and draw other types of graphs such as time-resolved histograms, per-residue fluctuations when compared to the first ($D_1$) and previous structure ($D_{-1}$) within the trajectory, etc.

### In-script usage III: creating custom graphs

Other types of graphs can be easily created by modifying part three of the code above. For example, the following code creates histograms of *R*, one for each model (starting from line 11 above).

```python
for chain in grouped_data.keys():                                                    11
    models, residues, Rs = grouped_data[chain]                                       12
                                                                                     13
    'Begin custom code'                                                              14
    X = []; Y=[]; Z=[]; # Will set X=model, Y=R, Z=P(R)                              15
    # Bundling the three lists into one 2d array                                     16
    new_data =  np.array(list(zip(models , residues , Rs)))                          17
    # Getting all R values, model by model                                          18
    for m in sorted(set(new_data[:,0])): # column 0 is the model column             19
        # Getting all Rs for that model #                                           20
        current_rs = new_data[np.where(new_data[:,0]==m)][:,2] # column 2 contains R 21
        # Getting the histogram                                                      22
        a,b = np.histogram(current_rs,bins=np.arange(0,1.01,0.01))                   23
        max_count = float(np.max(a))                                                 24
        for i in range(len(a)):                                                      25
            X.append(m); Y.append((b[i]+b[i+1])/2.0); Z.append(a[i]/float(np.sum(a)));  26
    'End custom code'                                                               27
                                                                                     28
    # Finally, creating (but not showing) the graph                                 29
    backmap.draw_xyz(X = X          ,        Y = Y   ,                Z = Z          30
            ,xlabel ='Frame #', ylabel ="R",zlabel ="P'(R)"                         31
            ,cmap = 'Greys', ylim=[0,1])                                            32
    plt.yticks(np.arange(0,1.00001,0.2))                                           33
    # Now, we display the graph:                                                   34
    plt.show() # ... one can also use plt.savefig() to save to file                35
```

The code above results in Fig. 9B.

### In-script usage IV: available color schemes (CMAPs)

Aside from the general color maps (cmaps) that exist in matplotlib (e.g., "Greys," "Reds," or, god forbid, "jet"), BACKMAP provides two new colormaps: "Chirality" (key: +ve twists: red; −ve twists: blue), and "SecondaryStructure" (key: *potential* α-helices: red; β-sheets: blue; ppII-helices: cyan). Right twisting backbones are shown in red; left twisting backbones are shown in blue. Figure 10 shows how a single protein ensemble may be described using these schematics. As illustrated in Fig. 10B, cmaps available within the standard matplotlib package do not distinguish between major secondary structures well, while those provided by BACKMAP do. In case it is known that the protein backbone

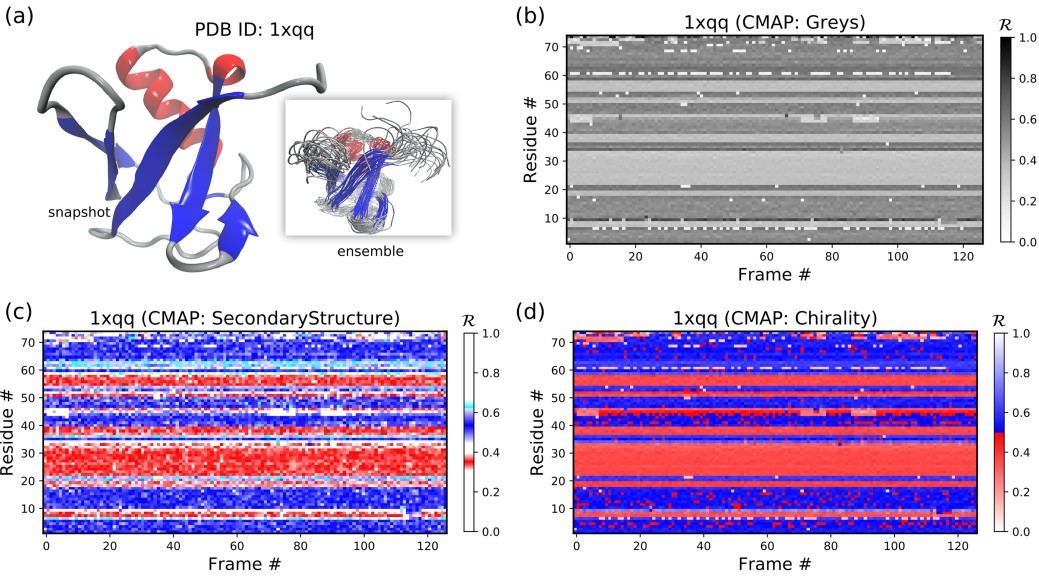

**Figure 10 A protein ensemble (A) along with some MAPs colored with different themes (B–D).** (C) and (D) are provided by the BACKMAP module. In (C), β-sheets are shown in blue and all helices are shown in red. In (D), right-handed and left-handed backbone twists are shown as red and blue, respectively.

accesses non-traditional regions of the Ramachandran plot, a four-color schematic will be needed (see below for more discussions).

## Stand alone usage

BACKMAP can be used as a stand alone package by running "> python -m backmap -pdb <pdb_dir_or_file>." The sectons below describe the expected outputs and how they may be interpreted.

### Stand alone Example I: a stable protein

Figures 11B–11F below were created by running "> python -m backmap ./tests/pdbs/1xqq. pdb" (Fig. 11A was created using VMD). These graphs indicate that conformational ensemble 1xqq describes a conformationally stable protein, since 1) each conformation shows little change in the $\mathcal{R}$ histogram over time (Fig. 11B), and 2) each residue fluctuates little in color (structure) over "time" (Figs. 11C–11F; see Methods). Here and below, it is assumed that discrete models represent distinct states of the protein over "time".

In particular, each column in Fig. 11B describes the histogram in Ramachandran number ($\mathcal{R}$) space for a single model/timeframe. These histograms show the presence of both α-helices (at $\mathcal{R} \approx 0.34$) and β-sheets (at $\mathcal{R} \approx 0.52$). Additionally, Figs. 11C and 11D describe per-residue conformational plots (colored by two different metrics or CMAPs), which show that most of the protein backbone remains relatively stable over time (e.g., few fluctuations in state or "color" are evident over frame #). Finally, Fig. 11E describes the extent to which a single residue's state has deviated from the first frame, and Fig. 11F describes the extent to which a single residue's state has deviated from its state in the previous frame. All these graphs show that this protein is relatively conformationally stable.
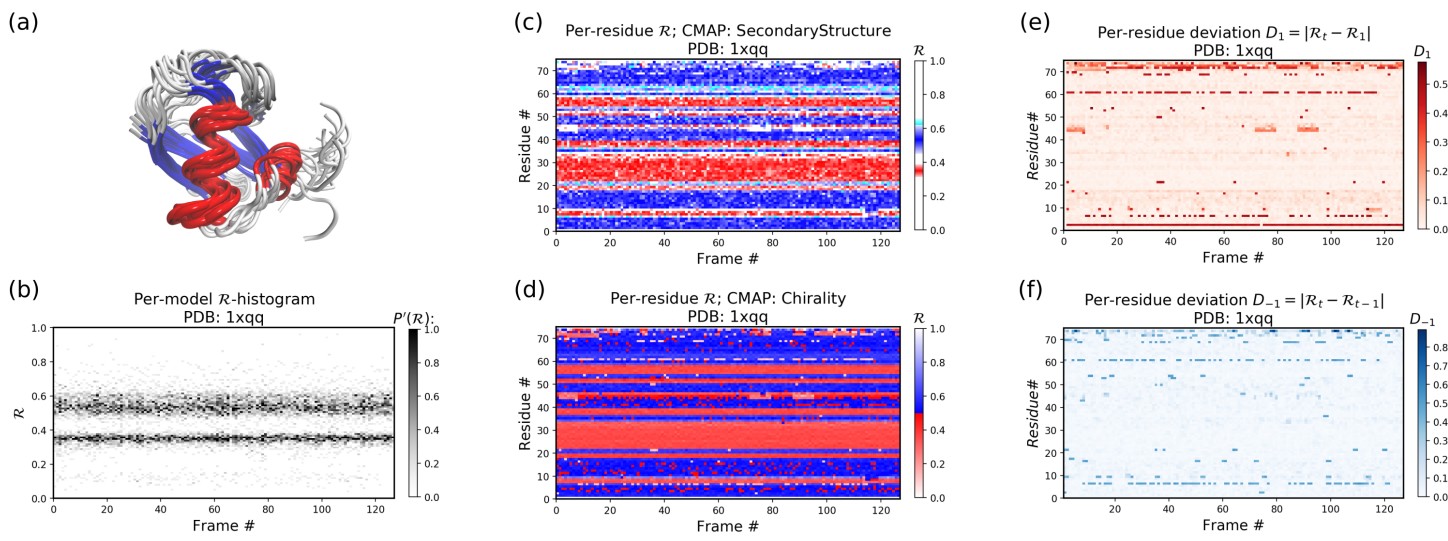

**Figure 11 Protein 1xqq describes a stable protein.** (A) represents the entire ensemble, (B) represents a histogram distribution of $\mathcal{R}$. (C) and (D) represent two ways to color per-residue $\mathcal{R}$ plots, and (E) and (F) are two ways to describe backbone fluctuation over time.

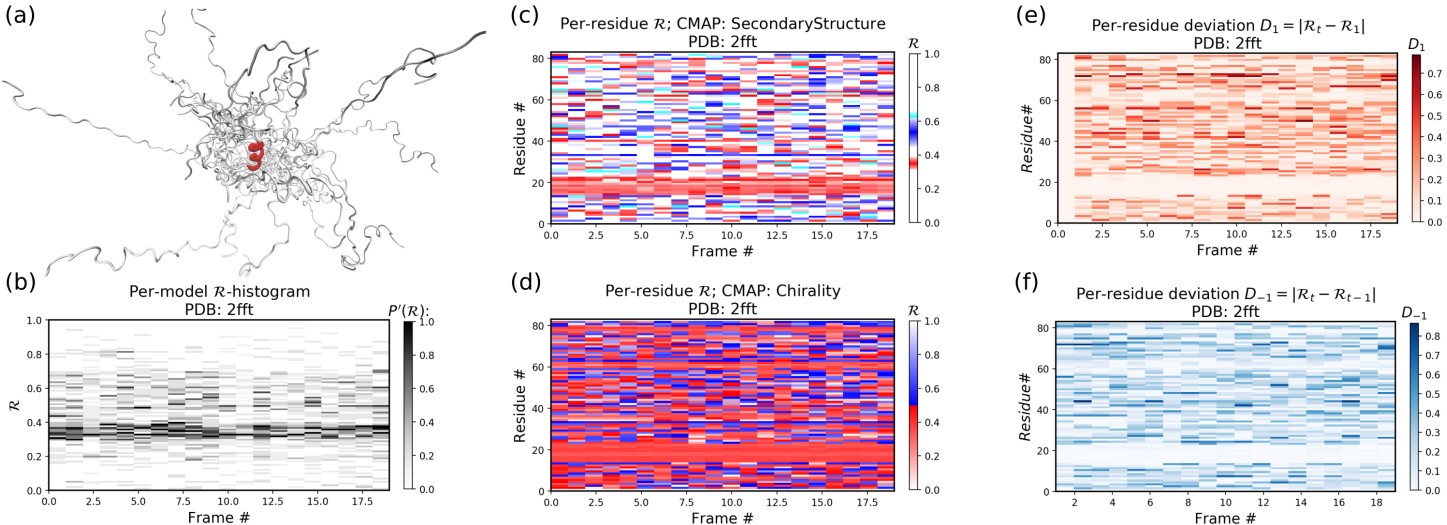

**Figure 12 Protein 2fft describes an intrinsically disordered protein.** While most of the backbone is intrinsically disordered, one region—a stable helix—is shown in red (A). (A) represents the entire ensemble, (B) represents a histogram distribution of $\mathcal{R}$. (C) and (D) represent two ways to color per-residue $\mathcal{R}$ plots, and (E) and (F) are two ways to describe backbone fluctuation over time.

### Stand alone Example II: an intrinsically disrodered protein

Figure 12 is identical to Fig. 11, except that the panels pertain to an intrinsically disordered protein 2fft whose structural ensemble describes dramatically distinct conformations.

As compared to the conformationally stable protein above, protein 2fft is much more flexible. Figure 11B shows that the states accessed per model are diverse and dramatically fluctuate over the entire range of $\mathcal{R}$ (this is especially true when compared to a stable protein, see Fig. 11B).

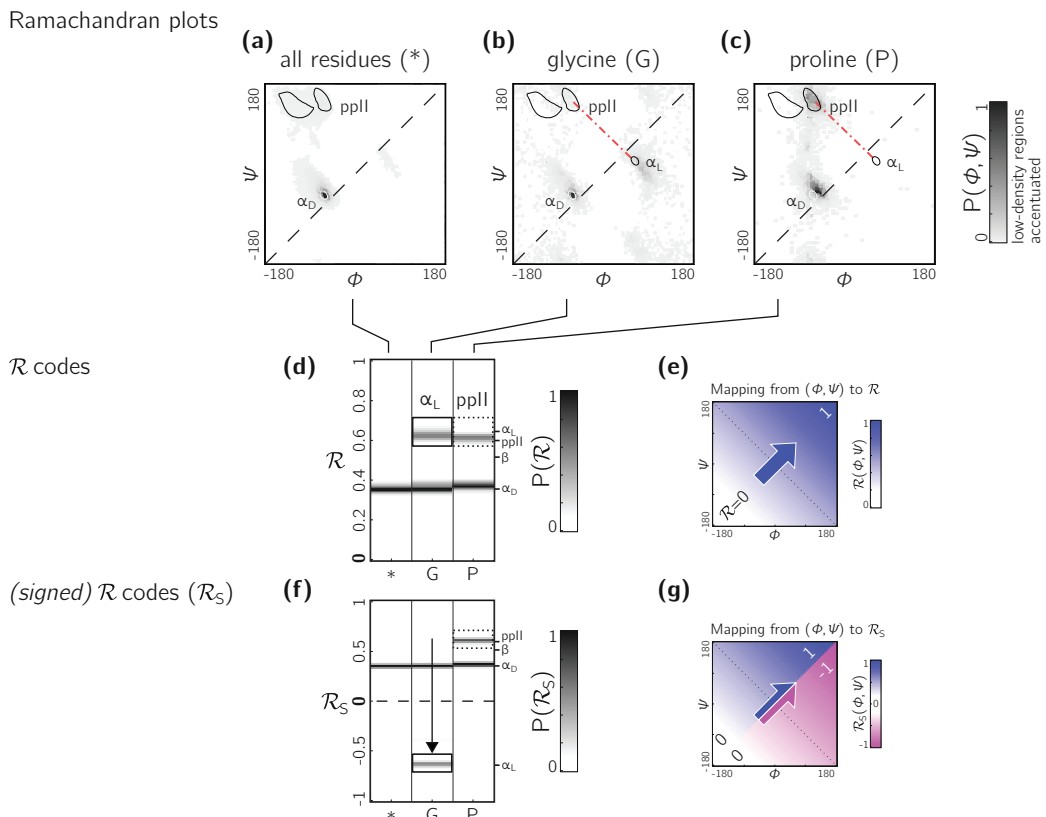

**Figure 13 Signed $\mathcal{R}_s$ are required for non-chiral backbones.** (A), (B) and (C) describe histogram distributions in $(\phi, \psi)$-space for all residues, glycine, and proline, respectively. (D) shows the same distributions, but as $\mathcal{R}$-codes. While the backbones of most amino acids occupy the top of the positively sloped diagonal (dashed in a), non-chiral amino acids such as Glycines (or their N-substituted variants—peptoids) display no strong preference, which causes distinct secondary structures that lie on the same "sweep" to be localized at similar regions in $\mathcal{R}$ (e.g., in d, polyproline-II and $\alpha_D$ helices both localize at $\mathcal{R} \approx 0.6$). However, a signed Ramachandran number ($\mathcal{R}_S$) solves this problem by multiplying those $\mathcal{R}$'s derived from backbones with $\phi > \psi$ by $-1$. The resolving power of $\mathcal{R}_S$ is evident by the separation of polyproline-II and $\alpha_D$ helices (F). The mapping of $(\phi, \psi)$ to $\mathcal{R}$ and $\mathcal{R}_S$ are respectively, shown in (E) and (G).

The diverse states occupied by each residue (Figs. 11C and 11D) confirm the conformational variation displayed by most of the backbone (Figs. 11E and 11F similarly show that most of the residues fluctuate dramatically).

Yet, interestingly, Figs. 11C–11F also show an unusually stable region—residues 15–25—which consistently display the same conformational ($\alpha$-helical) state at $\mathcal{R} \approx 0.34$ (interpreted as the color red in Fig. 11C). This trend would be hard to recognize by simply looking at the structural ensemble (Fig. 11A).

## A signed Ramachandran number for "misbehaving" backbones

The Ramachandran number increases in value from the bottom left of the Ramachandran plot to the top right in sweeps that are parallel to the negative sloping diagonal. As discussed in *Mannige, Kundu & Whitelam (2016)*, this method of mapping a two-dimensional space into one number is still structurally meaningful and descriptive

because (1) most structural features of the protein backbone—for example, radius of gyration (*Mannige, Kundu & Whitelam, 2016*), end-to-end distance (*Mannige, Kundu & Whitelam, 2016*), and chirality (*Mannige, 2017*)—vary little along lines parallel to the negatively-sloping diagonal (this is indicated by relatively small standard deviations in structural metrics for similar $\mathcal{R}_s$; Fig. 4), and (2) most protein backbones display chiral centers and therefore predominantly appear on the top left region of the Ramachandran plot (above the dashed diagonal in Fig. 13A).

However, not all backbones localize in only one half of the Ramachandran plot. Particularly, among biologically relevant amino acids, glycine occupies both regions of the Ramachandran plot (Fig. 13B; of note, the $\alpha_L$ helix region becomes relatively prominent). On the other hand, prolines are known to form polyproline-II helices (ppII in Fig. 13C), which falls on almost the same "sweep" as glycine rich peptides (red dot-dashed line). In situations where both prolines and glycines are abundant, the Ramachandran number ($\mathcal{R}$) would fail to distinguish $\alpha_L$ from ppII (Fig. 13D; regions outlined by rectangles).

To accomodate the situation where achiral backbones are expected (e.g., if peptoids or polyglycines are being studied), an additional Ramachandran number—the *signed* Ramachandran number $\mathcal{R}_s$—is introduced here. $\mathcal{R}_s$ is identical to the original number in magnitude, but which changes sign from + to − as you approach $\mathcal{R}$ numbers that are to the right (or below) the positively sloped diagonal i.e.,

$$\mathcal{R}_S = \begin{cases} \mathcal{R} & , \text{if } \psi \geq \phi \\ \mathcal{R} \times -1 & , \text{if } \psi < \phi \end{cases} \qquad (3)$$

Signed Ramachandran numbers are calculated by adding "`-signed`" within the command line implementation or by adding "`signed=True`" when making `backmap.R()` calls.

As an example of the utility of $\mathcal{R}_s$, Fig. 13F shows that $\mathcal{R}_s$ easily distinguishes $\alpha_D$ from ppII.

Note that the signed $\mathcal{R}_s$, while useful, would be important in very limited scenarios, as more than 96% of the amino acids in the PDB occupy the upper-left region of the Ramachandran plot (with 3% of "rule breakers" contributed mostly by glycines).

## CONCLUSION

A simpler Ramachandran number is reported—$\mathcal{R} = (\phi + \psi + 2\pi)/(4\pi)$—which, while being a single number, provides much information. For example, for proteins within the protein databank, $\mathcal{R}$ values above 0.5 are left-handed in twist, while those below 0.5 are right handed; $\mathcal{R}$ values close to 0, 0.5, and 1 are extended; β-sheets occupy $\mathcal{R}$ values at around 0.52; and right-handed α-helices hover around 0.34. Given the Ramachandran number's "stackability," single graphs can hold detailed information of the progression/evolution of molecular trajectories. Indeed, Fig. 8 shows how 400 distinct Ramachandran plots can easily be fit into one graph when using $\mathcal{R}$. Finally, a python script/module (BACKMAP) has been provided in an online GitHub repository to promote the utility of $\mathcal{R}$ as a universal metric.

## MATERIALS

Statistics about single amino acid conformations and secondary structures (excepting ppII) were derived from the Structural Classification of Proteins or SCOPe website (Release 2.06; *Fox, Brenner & Chandonia, 2014*). This database, currently available at http://scop.berkeley.edu/downloads/pdbstyle/pdbstyle-sel-gs-bib-40-2.06.tgz, contains 13,760 three-dimensional protein conformations (one domain per conformation) with lower than 40% sequence identity. Secondary structure annotations were assigned using the DSSP algorithm (*Kabsch & Sander, 1983*), although the STRIDE algorithm (*Frishman & Argos, 1995*) provides qualitatively identical distributions. These statistics were used to produce distributions within Figs. 2A, 3A and 3C.

Given the absence of ppII annotation in the present version of DSSP, statistics for ppII (used to generate the ppII distributions in Figs. 2A, 3A and 3C) were obtained from segments within 16,535 proteins annotated by PolyprOnline (*Chebrek et al., 2014*) to contain three or more residues of the secondary structure.

Figure 6 represents a trajectory of a portion of a single peptoid backbone within a "relaxing" peptoid nanosheet bilayer. The conformation of this backbone—derived from work by *Mannige et al. (2015)* and *Mannige, Kundu & Whitelam (2016)*—is also available as "/tests/pdbs/nanosheet_birth_U7.pdb" within the companion GitHub repository.

The following protein structures were obtained from the PDB: 1mba, 2acy, 1xqq, and 2fft. The first two in the list (1mba, 2acy) describe single conformations and the last two (1xqq, 2fft) describe ensembles. $\mathcal{R}$-based MAPs were created for each structure $X \in [nanosheet\_birth\_U7.pdb, 2fft, 2acy, 1xqq, 1mba]$ using the following command line code:

```
> python -m backmap -pdb tests/pdbs/X.pdb
```

The output of this command line implementation were used in Figs. 5B, 6B and 10B, 11B and 12B.

In order to describe changes in structure, this report uses two metrics for structural deviation: deviation in present structure when compared to the first conformation in the trajctory ($D_1$), and deviation in structure compared to the previous conformation in the trajctory ($D_{-1}$). For any residue $r$ at time $t$, these equations can be described as follows:

$$D_1 = |\mathcal{R}_t - \mathcal{R}_1|, \qquad D_{-1} = |\mathcal{R}_t - \mathcal{R}_{t-1}|. \tag{4}$$

All three-dimensional representations of proteins (Figs. 5A, 6A and 10A, 11A and 12A) were created using VMD (*Humphrey, Dalke & Schulten, 1996*). Finally, all other figures—excepting Fig. 1 that is derived from *Mannige, Kundu & Whitelam (2016)*—were created using helper Python scripts available in manuscript/python_generators/ within the companion GitHub repository.

## APPENDIX

### Simplifying the Ramachandran number ($\mathcal{R}$)

This section will derive the simplified Ramachandran number presented in this paper from the more complicated looking Ramachandran number introduced previously (*Mannige, Kundu & Whitelam, 2016*).

Assuming the bounds $\phi \in [\phi_{\min}, \phi_{\max})$ and $\phi \in [\psi_{\min}, \psi_{\max})$, the previously described Ramachandran number takes the form

$$\mathcal{R}(\phi, \psi) \equiv \frac{R_{\mathbb{Z}}(\phi, \psi) - R_{\mathbb{Z}}(\phi_{\min}, \phi_{\min})}{R_{\mathbb{Z}}(\phi_{\max}, \phi_{\max}) - R_{\mathbb{Z}}(\phi_{\min}, \phi_{\min})}, \tag{5}$$

where, $\mathcal{R}(\phi, \psi)$ is the Ramachanran number with range $(0, 1)$, and $R_{\mathbb{Z}}(\phi, \psi)$ is the *unnormalized* integer-spaced Ramachandran number whose closed form is

$$R_{\mathbb{Z}}(\phi, \psi) = \left[(\phi - \psi + \lambda)\sigma/\sqrt{2}\right] + \left[\sqrt{2}\lambda\sigma\right]\left[(\phi + \psi + \lambda)\sigma/\sqrt{2}\right]. \tag{6}$$

Here, $[x]$ rounds $x$ to the closest integer value, $\sigma$ is a scaling factor, discussed below, and $\lambda$ is the range of an angle in degrees (i.e., $\lambda = \phi_{\max} - \phi_{\min}$). Effectively, this equation does the following: (1) It divides up the Ramachandran plot into $(360° \ \sigma^{1/°})^2$ squares, where $\sigma$ is a user-selected scaling factor that is measured in reciprocal degrees (see Fig. 8B in *Mannige, Kundu & Whitelam (2016)*); (2) It then assigns integer values to each square by setting the lowest integer value to the bottom left of the Ramachandran plot ($\phi = -180°$, $\psi = -180°$), and proceeding from the bottom left to the top right of the Ramachandran plot by iteratively slicing down $-1/2$ sloped lines and assigning increasing integer values to each square that one encounters; (3) Finally, the equation assigns any $(\phi, \psi)$ pair within $\phi, \psi \in [-\phi_{\min}, \phi_{\max})$ to the integer value ($R_{\mathbb{Z}}$) assigned to the divvied-up square that they it exists in. Further discussions on this process are available in the previous publication (*Mannige, Kundu & Whitelam, 2016*).

Combining the two equations (Eqs. (5) and (6)) results in the following, rather imposing, equation for the Ramachandran number:

$$\mathcal{R}(\phi, \psi) = \frac{\begin{pmatrix} \left[(\phi - \psi + \lambda)\sigma/\sqrt{2}\right] & +\left[\sqrt{2}\lambda\sigma\right]\left[(\phi + \psi + \lambda)\sigma/\sqrt{2}\right] \\ -\left[(\phi_{\min} - \psi_{\min} + \lambda)\sigma/\sqrt{2}\right] & -\left[\sqrt{2}\lambda\sigma\right]\left[(\phi_{\min} + \psi_{\min} + \lambda)\sigma/\sqrt{2}\right] \end{pmatrix}}{\begin{pmatrix} \left[(\phi_{\max} - \psi_{\max} + \lambda)\sigma/\sqrt{2}\right] & +\left[\sqrt{2}\lambda\sigma\right]\left[(\phi_{\max} + \psi_{\max} + \lambda)\sigma/\sqrt{2}\right] \\ -\left[(\phi_{\min} - \psi_{\min} + \lambda)\sigma/\sqrt{2}\right] & -\left[\sqrt{2}\lambda\sigma\right]\left[(\phi_{\min} + \psi_{\min} + \lambda)\sigma/\sqrt{2}\right] \end{pmatrix}} \tag{7}$$

However, useful Eq. (7) is, the complexity of the equation may be a deterrent toward utilizing it. This paper reports a simpler equation that is derived by taking the limit of Eq. (7) as $\sigma$ tends toward $\infty$. In particular, when $\sigma \to \infty$, Eq. (7) becomes

$$\mathcal{R}(\phi, \psi) = \lim_{\sigma \to \infty} \bar{\mathcal{R}}(\phi, \psi) = \frac{\phi + \psi - (\psi_{\min} + \psi_{\min})}{(\phi_{\max} + \psi_{\max}) - (\phi_{\min} + \psi_{\min})}. \tag{8}$$

Assuming that $\phi, \psi \in [-180°, 180°)$ or $[-\pi, \pi)$,

$$\mathcal{R}(\phi, \psi) = \frac{\phi + \psi + 2\pi}{4\pi}. \tag{9}$$

Conformation of this limit is shown numerically in Fig. A1. Since larger $\sigma$s indicate higher accuracy, $\lim_{\sigma \to \infty} \mathcal{R}(\phi, \psi)$ represents an exact representation of the Ramachandran number. Using this closed form, this report shows how both static structural features and complex structural transitions may be identified with the help of Ramachandran number-derived plots.

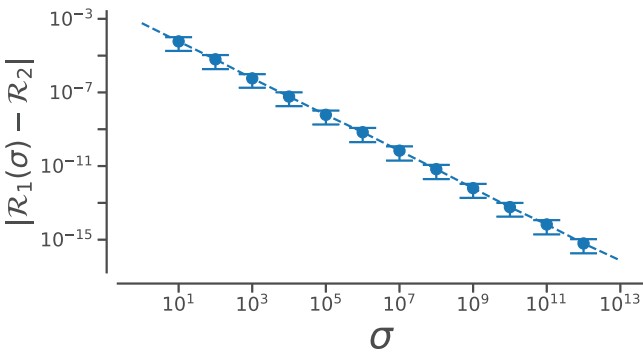

**Figure A1 Relating old $\mathcal{R}_1$ and new $\mathcal{R}_2$ closed forms that describe $\mathcal{R}$.** The increase in the accuracy measure ($\sigma$) for the original Ramachandran number (Eq. (6)) results in values that tend toward the new Ramachandran number proposed in this paper (Eq. (2)).

Assuming, a different range (say, $\phi, \psi \in [0, 2\pi)$), the Ramachandran number in that frame of reference will be

$$\mathcal{R}(\phi, \psi)_{\phi,\psi \in [0,2\pi)} = \frac{\phi + \psi}{4\pi}. \tag{10}$$

However, in changing the ranges, the meaning of the Ramachandran number will change. This manuscript assumes that all angles ($\phi, \psi, \omega$) range between $-\pi$ ($-180°$) and $\pi$ ($180°$).

## ACKNOWLEDGEMENTS

RVM thanks Alana Canfield Mannige for her input. The notion of the signed Ramachandran number emerged from discussions with Joyjit Kundu and Stephen Whitelam while at the Molecular Foundry at Lawrence Berkeley National Laboratory (LBNL).

### Funding

During the development of this paper, Ranjan Mannige was partially supported by the Defense Threat Reduction Agency under contract No. IACRO-B0845281. This work was partially done at the Molecular Foundry at LBNL, supported by the Office of Science, Office of Basic Energy Sciences, of the U.S. Department of Energy under Contract No. DE-AC02-05CH11231. There was no additional external funding received for this study. The funders had no role in study design, data collection and analysis, decision to publish, or preparation of the manuscript.

### Grant Disclosures

The following grant information was disclosed by the authors:
Defense Threat Reduction Agency under Contract No: IACRO-B0845281.
Molecular Foundry at LBNL, supported by the Office of Science, Office of Basic Energy Sciences, of the U.S. Department of Energy under Contract No: DE-AC02-05CH11231.

# PeerJ

## Competing Interests

The authors declare that they have no competing interests.

## Author Contributions

- Ranjan Mannige conceived and designed the experiments, performed the experiments, analyzed the data, contributed reagents/materials/analysis tools, prepared figures and/or tables, authored or reviewed drafts of the paper, approved the final draft, developed python module.

## Data Availability

All data and figures were generated using the Python scripts available in the following public GitHub repository: https://github.com/ranjanmannige/backmap.

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
