# Peer review of "The BackMAP Python module: how a simpler Ramachandran number can simplify the life of a protein simulator"

_PeerJ, doi:10.7717/peerj.5745_

## Round 0.1 · original submission · Minor Revisions

Please address all critical issues raised by both reviewers and revise your manuscript accordingly.

·

Basic reporting

This article is well written in generally clear language. The author should be commended for his clarity. The introduction and background effectively provide context and motivation for the work.

A new piece of software (BackMAP) is a major part of the paper, and it is available and works out of the box in my hands, which is great.

The included references are reasonable. However, the author references himself quite frequently (which is related to some weakness of the paper – see below), and omits some salient papers on Ramachandran plots. For example:
• Lovell et al. Proteins (2003): high-quality Rama plots + smoothed distributions
• Ting et al. PLoS Comp Biol (2010): more general exploration of neighbor-dependent Rama plots

The figures are generally clear and quite aesthetically pleasing, which is a credit to the new software package, which was used to generate many of the figure panels. However, there are a few weaknesses:
• Several figures do not have a caption, which makes them more difficult to interpret. Please add captions.
• Other figures have insufficient captions. For example, in the Figure 2 caption, it is not clear what the dots in panel b are, nor what the different colors mean in b-c (presumably for emphasis, but this is not stated).
• Some figures are incorrectly referenced in the text and captions. For example, in the Figure 2 caption, I think “qualitative sense of peptide backbone structure (a, c), it is not a convenient representation for exploring peptide backbone dynamics (c)” should be “qualitative sense of peptide backbone structure (a, b), it is not a convenient representation for exploring peptide backbone dynamics (c)”.
• Why are Figure 8c-d so “dotty” as compared to all other MAP images? Describe this in the caption?

Although the messaging and language of the manuscript are generally clear, they could be improved in places:
• In e.g. lines 116 onward, “chirality” is discussed with respect to protein backbone. Typically this means L vs. D amino acids, i.e. which direction the sidechain diverges from the backbone. Here the definition is something different, but is not clearly defined. It seems to mean a property that would be observed if several sequential residues were to adopt the same conformation. This idea was apparently explored more in the author’s earlier papers on this topic, but is not made sufficiently clear here for this standalone paper, which is a repeated theme.
• Similarly, peptoids are discussed in the “Ramachandran codes are stackable” section, but it is not made clear how they differ from peptides. Peptoids were previously studied by the author.
• Lines 88-91: In this overly terse section, the residue-level parameter R is related to several protein-level parameters, but it is not made clear in the text how such a mapping makes any sense – this is only described in the middle of the associated figure’s caption. Please (briefly) clarify this in the main text.
• Something called “MAPs” are introduced in line 73, but they are not defined until much later. Please (briefly) define when first introduced.
• Lines 38-39: “Today, protein structures described in context of the two-dimensional (φ , ψ )-space are called Ramachandran plots.” This is imprecise: the structures themselves are not called Rama plots; the structures are described by the plots.
• Figures 7 and 8 are titled “Ramachandran lines are stackable”, yet these figures are not referenced in the text section called “Ramachandran codes are stackable”, which cites a different figure… This is unnecessarily confusing for the reader.
• Abstract: “closed form” vs. “closed-form” – be consistent (or maybe “closed-form expression”)
• Lines 263-264: Explain how these quantities are defined (and see below for suggestions regarding RMSD/RMSF)
• This paper uses RMSD and RMSF here very differently from the usual definitions. Here, both RMSD and RMSF are defined as a difference in two R numbers, squared, and square-rooted. Usually they are sums of many such squared terms that are then square-rooted. Without a sum, an absolute value operator is sufficient. Moreover, RMSF typically refers to deviations from an average, but here it simply refers to deviations from the previous entry in the sequence. Please simplify these equations by switching to absolute value operators and change the nomenclature to avoid overloading these terms.
• How were the “Stand Alone Example” structures obtained? Mention the experimental (or computational) method.
• Line 81: “most” -> “more”

Generally, there are several careless errors scattered through the manuscript. I list a few of them here:
• Lines 320 & 323: “Fig X” = placeholder?
• Figure 12 caption – typo: “evident available by”
• Line 327: typo
• Last line: missing period
• Line 81: missing Section #
• Line “254: Panel (b)” should be “Panel (a)”
• Figure 7 caption: “Ramachanran”
• (maybe more)

Experimental design

The experimental design, so to speak, of the paper is solid. The paper provides several relevant examples that bridge Rama plots, the notion of R, multiple-model ensembles, different types of ensembles, and several possible bioinformatics applications. The paper is relatively heavily focused on a new software package, so naturally a good portion of it is dedicated to that rather than “experimental design” per se.

Validity of the findings

The findings and discussion of the paper are generally valid, with a couple of exceptions:

• I have read the section “Ramachandran numbers are more compact than one might realize” several times, and still think it makes no sense. The paper states that “for any N-length peptide, any ordering of [φ1,φ2,...,φN,ψ1,ψ2,...,ψN] can not describe the structure”. I don’t see how this is true (at least for the backbone structure alone, and ignoring bond lengths and angles, as I assume the author is doing)… A series of φ and ψ angles certainly CAN define a backbone conformation in this sense… I’m not sure what is meant by “any ordering” – of course a random order would specify the wrong conformation – but I don’t think that’s what the author is going for… Anyway, please either clarify this or (more likely) just remove this section – the point is already made that the dimensionality reduction of R can be useful!
• In the section “Case study: picking out subtle differences from high volume of data” and Figure 8, it is stated that pre-Pro has distinct conformational preferences, which makes sense. But it is also stated that pre-Gly has distinct conformational preferences, which is not well supported by the data shown. It is, however, clear that Gly residues themselves have distinct conformational preferences. The references included on line 142 seem to address different issues: Gly and Pro positions within helices, and energetics of Gly and pre-Pro. (See above, however, for other references to include.)

Additional comments

This manuscript discusses a previously introduced parameter called the Ramachandran number (R), which effectively reduces a 2D Ramachandran plot to a 1D variable. This seems difficult initially since the 2D Ramachandran plot is rich with features, but the paper does a reasonable job of alleviating those concerns, largely since the major features of protein structures in the Rama plot (perhaps fortuitously) do not directly overlap in R space. R thus may be generally useful for structural bioinformatics. The paper surveys some potential uses of this number, and offers a software package which makes it easy to use for the reader to explore structures of interest to them – in particular ensembles and dynamic trajectories, which is a forward-looking perspective.

My main criticism is that, in contrast to the majority of the paper, a few aspects of the work are not well explained. These aspects seem to correspond to aspects the author has explored in his previous papers already – it seems he does not want to repeat himself. However, without these inclusions, the paper has readability gaps. Most notably, how does the Ramachandran number R map onto a Ramachandran plot? A simple heatmap of R in a 2D Rama plot as a Figure 1 would help build intuition quickly, which would aid the reader in digesting the rest of the paper. Only much later, lines 280-281 say “The Ramachandran number increases in value from the bottom left of the Ramachandran plot to the top right in sweeps that are parallel to the negative sloping diagonal.” The reader could of course derive this him- or herself based on the equation, but since R is so central to the entire paper, making sure to establish this intuition up-front is important. Please move this to the very beginning when R is introduced! Other similar aspects, where the paper is sparse on explanation for topics the author has previously published on, are described elsewhere in the review (peptoids, chirality).

Also, while the use of the new(ish) R metric may be useful for many applications, the results presented here are generally retrospective. It is argued that these findings (e.g. that Gly, Pro, and pre-Pro residues have unique conformational preferences) would be difficult to make without R – but in fact they were made without R. This is not necessarily a knock against the method, but rather more of a validation that sets up the method to be used for other prospective applications by others later. However, the language in the paper could be toned down a notch to emphasize these observations were made previously. (As noted above, some references are included, but other salient ones are not.)

Overall, this paper is a useful contribution that may attract many readers interested in structural bioinformatics and other aspects of computational structural biology.

·

Basic reporting

The paper is well written and clear, but there are abundant typos that need to be fixed. (several are highlighted on attached PDF)

The background is sufficient, although the references are on the light side. A glaring example was a single reference (by the author) supporting the existence of conformational changes.

The figures are fine, but several of the captions are inadequate to describe the content: Fig 2 (explain panels), 6, and 8 (panel d not mentioned).

Database files, code, and scripts are provided.

Experimental design

The goals and methods are well defined and competently carried out.

Validity of the findings

The advancement over the previous publication is subtle, but the author makes a strong case for its inclusion in the literature. It is well suited for this journal.

Additional comments

I am unconvinced by the argument (line 92-102) that the pairing of dihedral angles somehow increases the complexity of the 2N numbers required to specify a backbone configuration. Perhaps the author is referring to the limited region of the Ramachandran plot that is available? But, there are always steric constraints on a polymer, and this leads to coupling over much greater distances along the backbone (Flory swelling). I would argue the other way: 2N numbers are required to specify the backbone, but because so much of the region is sterically forbidden, 2N is greatly in excess of what is actually needed.
The “MAP” acronym is used several times before it is defined.
On line 254 I think he means panel a instead of b.
On line 295 I think the reference should be to panel c.
On line 302 he discusses the “handedness” of amino acids. Based on the following discussion, I think he is referring to the handedness of a helix that would occur if the specified amino acid were propagated, but this could use clarification.

---

## Round 0.2 · accepted · Accept

All the critical points were adequately addressed and the manuscript was revised accordingly. Therefore, it can be accepted now.

#